# Urine lactate concentration as a non-invasive screener for metabolic abnormalities: Findings in children with autism spectrum disorder and regression

Sofie Boterberg[1]*, Elise Vantroys[2], Boel De Paepe[2], Rudy Van Coster[2‡], Herbert Roeyers[1‡]

1 Faculty of Psychology and Educational Sciences, Department of Experimental Clinical and Health Psychology, Research in Developmental Disorders Lab, Ghent University, Ghent, Belgium, 2 Faculty of Medicine and Health Sciences, Department of Internal Medicine and Paediatrics, Ghent University, Ghent, Belgium

‡ RVC and HR authors are joint senior authors on this work.
* sofie.boterberg@ugent.be

**Data Availability Statement:** Data cannot be shared publicly because they contain potentially

## Abstract

There is increasing evidence that diseases caused by dysfunctional mitochondria (MD) are associated with autism spectrum disorder (ASD). A comprehensive meta-analysis showed that developmental regression was reported in half of the children with ASD and mitochondrial dysfunction which is much higher than in the general population of ASD. The aim of the present exploratory study was to determine lactate concentrations in urine of children with ASD, as a non-invasive large-scale screening method for metabolic abnormalities including mitochondrial dysfunction and its possible association with regression. First, clinical characteristics of MD were examined in 99 children (3–11 years) with ASD. Second, clinical characteristics of MD, severity of ASD and reported regression were compared between children with the 20% lowest lactate concentrations and those with the 20% highest lactate concentrations in urine. Third, clinical characteristics of MD and lactate concentration in urine were compared in children with ($n = 37$) and without ($n = 62$) reported regression. An association of urine lactate concentrations with mitochondrial dysfunction and regression could not be demonstrated in our large ASD cohort. However, since ASD children were reported by their parents to show a broad range of phenotypic characteristics of MD (e.g., gastro-intestinal and respiratory impairments), and lactate concentrations in urine are not always increased in individuals with MD, the presence of milder mitochondrial dysfunction cannot be excluded. Development of alternative biomarkers and their implementation in prospective studies following developmental trajectories of infants at elevated likelihood for ASD will be needed in the future to further unravel the association of ASD with mitochondrial dysfunction and eventually improve early detection.

identifying or sensitive, clinical (minor – under 18 years of age) patient information. These restrictions are imposed by the ethical board of the Faculty of Psychology and Educational Sciences of The University Ghent and the University Hospital Ghent. Because of legal and ethical restrictions, it was not possible and appropriate to deposit the data in an external repository. Therefore, the data underlying the results presented in the study are only available from the corresponding author (sofie.boterberg@ugent.be) or by contacting the Department of Experimental Clinical and Health Psychology (secpp05@ugent.be).

**Funding:** SB is supported by the Research Foundation – Flanders; Belgium (FWO – 12Z8821N/ URL: https://www.fwo.be/). EV is supported by the Special Research Fund Belgium – Interdisciplinary Research Project (grant number: BOF.24J.2014.0005.02/ URL: https://www.ugent.be/en/research/funding/bof/iop). The funders had no role in study design, data collection and analysis, decision to publish, or preparation of the manuscript. There was no additional external funding received for this study.

**Competing interests:** The authors have declared that no competing interests exist.

**Abbreviations:** ASD, Autism Spectrum Disorder; GI, gastrointestinal; MD, mitochondrial diseases; ATP, adenosine triphosphate; OXPHOS, oxidative phosphorylation; mtDNA, mitochondrial DNA; nuDNA, nuclear DNA; MELAS, Mitochondrial Encephalomyopathy, Lactic Acidosis, and Stroke-like episodes; CSF, cerebrospinal fluid; DSM, Diagnostic and Statistical Manual of Mental Disorders; ADI-R, Autism Diagnostic Interview-Revised; RSQ, Regression Supplemental Questions; RVI, Regression Validity Interview; EDQ, Early Development Questionnaire; WNV, Wechsler Non-Verbal Scale of Ability; ADOS-2, The Autism Diagnostic Observation Scale–Second Edition; RRBs, restricted and repetitive behaviours; CSS, Calibrated Severity Scores; ASD-LL, group of children with ASD and the 20% Lowest Lactate concentrations; ASD-ML, group of children with ASD and the 60% Medium Lactate concentrations; ASD-HL, group of children with ASD and the 20% Highest Lactate concentrations; MMS, Mitochondrial Medicine Society; ASD-R, group of children with ASD and regression in language, social, motor and/or adaptive skills before, at, or after the age of 36 months not caused by brain injury or significant life event; ASD-R≤36M, children with ASD and regression before or at 36 months; ASD-R>36M, children with ASD and regression after 36 months; ASD-NR, group of children with ASD without regression; MCAR,

# Introduction

Autism spectrum disorder (ASD) is a common heterogeneous neurodevelopmental condition characterised by impairments in social communication and interaction and by restricted repetitive patterns of behaviour, interests and activities and sensory anomalies [1,2]. ASD has a global prevalence under 1% [3], but estimates are higher in high-resource countries [4]. Despite many recent scientific advances in the research of ASD, its pathogenesis remains largely unknown. However, different combinations of genetic and environmental risk factors as well as molecular abnormalities are likely to be involved [5–7]. Besides medical comorbidities such as epilepsy and seizures [8], sleep disruption [9] and gastrointestinal (GI) abnormalities [10], growing evidence indicates that a significant percentage of individuals with ASD actually have a metabolic disorder [11,12]. More specifically, mitochondrial diseases (MD), a large heterogeneous group of diseases caused by dysfunctional mitochondria, are increasingly being considered as possibly associated with ASD [11,13–18]. Studies have shown percentages of children with ASD suffering from concomitant MD (i.e., 5.0%; [11,12,19]) exceeding expected numbers, when compared to the prevalence of MD in the general population (i.e., 0.01–0.05%; [20,21]).

Mitochondria are cellular organelles which are regarded as the "powerhouse" of the cell as they are responsible for most of the cell's energy production, in the form of adenosine triphosphate (ATP) generated by the oxidative phosphorylation (OXPHOS) process. MD are some of the most common metabolic disorders in children and adolescents [22] mostly caused by defects in the OXPHOS system which consists of five enzyme complexes I-V, together with two electron carriers: ubiquinone and cytochrome c [23]. These complexes are composed of structural proteins encoded either by the mitochondrial genome (mtDNA) or by the nuclear genome (nuDNA). An example of MD caused by primary mutations in mtDNA is MELAS (Mitochondrial Encephalomyopathy, Lactic Acidosis, and Stroke-like episodes [24]). However, defects in nuDNA account for the vast majority of paediatric patients with mitochondrial dysfunction [22]. Further, mitochondrial dysfunction can be classified as primary (caused by a defect in a gene directly involved in the function of mitochondrial systems responsible for producing ATP) or secondary (caused by other metabolic or genetic abnormalities impairing mitochondrial production of ATP) [11,25].

The clinical presentation of MD is variable as different combinations of affected organs can occur and the severity of the defects varies from patient to patient and even between siblings carrying the same mutation. Defects in the OXPHOS system result in a decrease of ATP production, affecting most severely organs and tissues with high energy demand. Since the brain is the organ with the highest energy demand in the body, a mild mitochondrial defect in this organ will already have a significant impact on its function, while other organ systems will only be affected by more severe mitochondrial defects. Mild mitochondrial defects in the brain are suggested to be related with different neuropsychiatric disorders including ASD [26,27]. In childhood, apart from the neuromuscular system, many other organ systems can be affected such as heart, kidneys, liver, gastrointestinal tract, eyes and auditory system resulting in clinical presentations such as seizures, lethargy, hypotonia, developmental delay, cardiomyopathy, hearing or visual impairment and movement disorders [28,29]. Hence, diagnosis of MD is challenging because of the considerable clinical variability. Although some individuals diagnosed with MD show clinical features that fall into a defined clinical syndrome, many individuals do not [22].

Most diagnostic algorithms for MD recommend evaluation of a combination of mitochondrial biomarkers such as lactate and pyruvate concentrations in plasma and cerebrospinal fluid (CSF), plasma, urine, and CSF amino acids, plasma acylcarnitines, and urine organic acids

missing completely at random; SES, socioeconomic status; DCD, developmental coordination disorder; ADHD, attention deficit hyperactivity disorder; OCD, obsessive compulsive disorder; PWS, Prader-Willi syndrome; MRS, Magnetic Resonance Spectroscopy; NIRS, Near-Infrared Spectroscopy.

[30]. In case of compromised OXPHOS activity, pyruvate, the end product of glycolysis accumulates and is transformed into lactate in the cytoplasm leading to accumulation of lactate in blood and urine. Thus, measurement of concentration of lactate in blood and urine is routinely used as a first step in the screening process for MD in clinically suspected patients [30]. It was shown that when the blood lactate concentrations exceed a threshold (i.e., 3 mmol/l; [30]) the lactate concentrations in urine also markedly increase [31,32]. Further, when blood lactate concentration is only moderately increased (e.g., between 2.2 and 3 mmol/l), urinary lactate concentrations increase as well in most of the cases [32,33]. Only minimal amounts of lactate are excreted in the urine when the blood lactate concentrations are within normal range [31–34]. Thus, since lactate values in blood are correlated with lactate values in urine, urinary lactate concentration could be a useful non-invasive screening method for metabolic abnormalities including impaired mitochondrial functioning.

Based on nine different studies, the meta-analysis of Rossignol and Frye in 2012 demonstrated that about one-third of children with ASD had increased lactate concentration in blood [11]. When compared to typically developing controls, children with ASD showed significantly larger variability in the concentrations of both direct (e.g., lactate and pyruvate) and indirect (e.g., creatine kinase and carnitine) biochemical markers suggesting the presence of a 'spectrum' of mitochondrial dysfunction in children with ASD [11]. In this respect, mitochondrial dysfunction is suspected to be a prevalent feature of ASD rather than being present in only a discrete subgroup of children with ASD [11]. After the meta-analysis in 2012, increased concentrations of serum lactate continued to be reported in 66% [35], 80% [36] and 94% [37] of children with ASD.

Based on data from 150 children and adolescents with a combined diagnosis of ASD and MD (ASD+MD) up to the age of 20 years, Rossignol and Frye (2012) indicated that the prevalence of male gender, developmental regression, seizures, hypotonia, cardiomyopathy and myopathy did not differ between groups of MD children with and without ASD. Both groups only differed on the prevalence of fatigue/lethargy, ataxia, GI abnormalities and increased lactate, which was higher in the MD+ASD group as compared to the general population of MD [11]. When compared to the general ASD population, children with ASD+MD had more seizures (41%), motor delay (51%) and GI abnormalities (74%; including reflux and obstipation) [11,14]. Also, in 78% of the individuals in the comorbid group, lactate in blood was increased. Of interest is that developmental regression–or loss of previously acquired skills–in language, social interaction skills, play skills and motor skills was noticed in 52% of the children with ASD+MD, or with abnormal biochemical markers for mitochondrial function [11]. This is much higher than in the general population of ASD where two meta-analyses estimated regression prevalence at 30% [38,39]. Whether mitochondrial dysfunction contributed to or caused the reported regression in these children is unclear. However, patients with MD (without ASD) often present with regression [14] and evidence was found for mitochondrial dysfunction causing reduced synaptic neurotransmitter release in, for example, inhibitory GABAergic interneurons [40]. The latter play an important role in brain development between 12 and 30 months of age [41], which is the age range during which regression in ASD is most commonly reported [11,38,39].

Although several studies investigated a possible link between ASD and MD, the overall prevalence and specific role of metabolic abnormalities and mitochondrial dysfunction in ASD still remains undefined [11,12,14,19,42]. This could be due to small sample sizes and variety of screening procedures used [11,14].

The exploratory study presented here aimed to evaluate lactate concentration in urine of children with ASD, as a non-invasive large-scale screening method for metabolic abnormalities including mitochondrial dysfunction, and its possible association with regression. To the

best of our knowledge, the current study is the first to include (1) a large community-based sample, (2) a non-invasive technique to measure metabolic abnormalities, and (3) validated methods with regard to the measurement of ASD severity and developmental regression.

## Materials and methods

### Participants

This cross-sectional study was conducted in 99 children with ASD aged between 3 and 11 years old ($M$ = 7.55 years, $SD$ = 1.99; 72.7% boys). All children had an official community diagnosis of ASD confirmed by a multidisciplinary team fulfilling the criteria of the diagnosis of ASD according to the 4th (DSM-IV-TR; [43]) or 5th (DSM-5 [1]) edition of the Diagnostic and Statistical Manual of Mental Disorders (DSM). Initially, 103 children were recruited from a community-based sample via social media, parent associations of children with ASD, home guidance organizations and different multidisciplinary rehabilitation centres through online and newsletter advertisements. The primary purpose of the recruitment was to study the general and biological development of children with ASD (some children were also included in a related study [44]). However, in 3 children no urine samples were provided by the parents and in 1 child there was no official community diagnosis of ASD as defined above. Therefore these children ($n$ = 4) were excluded from the present analyses. For an overview of the participant selection process see also the recruitment and research flow chart in the (S1 File).

### Measures

**Biomarker lactate concentration in urine.**   Lactate was measured using the LACT2 protocol from Cobas ® Roche diagnostics GmbH as described by the manufacturer. As the children were young, a non-invasive technique to detect metabolic abnormalities by measuring lactate concentrations in urine samples was preferred to blood drawing. Another reason to prefer measurement of lactate in urine is that in previous studies false-positive increases of lactate in blood were found caused by difficulties during blood sampling (e.g., struggling of the child during the blood draw; [14,30,45]) rather than by actual mitochondrial dysfunction.

**Clinical characteristics of mitochondrial dysfunction.**   Two self-developed interviews by the authors (see also S2 File and S3 File) concerning associated clinical characteristics of mitochondrial diseases and dysfunctions in the child and the family were conducted with the parents. The questions in the interviews were based upon previous review studies on clinical characteristics and psychiatric disorders associated with mitochondrial dysfunction [11,14,26,28,29,46].

**History of developmental regression.**   The Autism Diagnostic Interview-Revised (ADI-R [47]) is a semi-structured, 111-item, diagnostic parent interview which is administered to classify ASD in children or adults. In the present study, information based on items #11–19 on history of language regression and items #20–28 on potential losses of other skills such as motor, self-help, play and social abilities was included. Additionally, a supplemental interview [a Dutch version of the Regression Supplemental Questions (RSQ [48]) derived from the Regression Validity Interview; RVI [49]] which captures information about more subtle skill losses in the social-communicative domain (e.g., eye contact, social smiling and waving goodbye) was implemented. To enhance the reliability of the parent report of regression using the interviews ADI-R and RSQ, a Dutch version of the Early Development Questionnaire (EDQ [50]), which collects retrospective information on characteristics of ASD onset, was added.

**Cognitive functioning.**   The Wechsler Non-Verbal Scale of Ability (WNV [51]) is a non-verbal measure of ability for children and adolescents aged 4 to 21 years, especially designed for children who have communicative disabilities or ASD. Since the WNV uses pictorial

directions for informing the child on the demands of the test, it makes it very suitable for individuals with ASD, as was shown in previous studies [52,53]. Based on the results of the subtests, a single ability score can be derived.

**ASD-characteristics and severity.** The Autism Diagnostic Observation Scale–Second Edition (ADOS-2 [54]) is a semi-structured, standardized assessment of communication and social interaction, play, and restricted and repetitive behaviours (RRBs). In line with previous research [55] Calibrated Severity Scores (CSS) were used for Social Affect, RRBs, and the Total Score to account for the variation in age, language level and module administration [56,57].

## Procedure

Each child was individually evaluated by one examiner with professional background as clinical psychologist and who had received training in assessment and interpretation of the tests. In the present study, all urine samples were collected postprandial (since in this way lactate concentration is more representative than fasting specimens; [30]) during the research setting or at home. The urine collected through a urine bag (in young children) or a urine sample beaker was stored in test tubes with a volume of 11 mL and immediately frozen at a temperature of -18˚ Celsius before analysis within maximum 3 months after collection. In the lab, after thawing, the samples were immediately analysed. The study was conducted under approval of the ethical board of the Faculty of Psychology and Educational Sciences of the University Ghent (Project number: 2015/51; date: 03/08/2015) and the University Hospital Ghent (Belgian registration number: B670201525767; Project number: EC UZG 2015/1023; date: 27/10/2015) as conforming to the Declaration of Helsinki, 2000. In addition, written informed consent was obtained from the parents or legal guardians of the children.

## Data analysis

**Group membership allocation based on lactate concentration in urine.** Based on the concentrations of lactate in urine ($M$ = 6.0 mg/dL, $SD$ = 3.6 mg/dL; range: .5–17.8 mg/dL), three groups could be distinguished: a group of children with the 20% Lowest Lactate concentrations (i.e., ASD-LL; $n$ = 20), a group of children with the 60% Medium Lactate concentrations (i.e., ASD-ML; $n$ = 59) and a group of children with the 20% Highest Lactate concentrations (i.e., ASD-HL; $n$ = 20). In total, five children showed an increased lactate concentration (range: 13.8–17.8 mg/dL) as compared to the other children (Fig 1, Panel A).

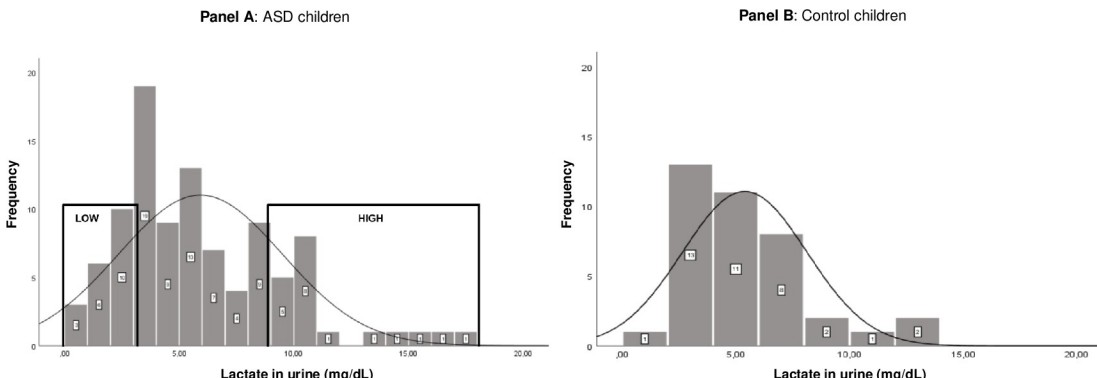

**Fig 1.** Distribution of lactate concentrations in children with ASD ($N$ = 99; Panel A) and control children ($N$ = 38; Panel B). Panel A: LOW = group of children with the 20% Lowest Lactate concentrations (i.e., ASD-LL; n = 20; range = .5–3.1 mg/dL) and HIGH = group of children with the 20% Highest Lactate concentrations (i.e., ASD-HL; n = 20; range = 8.9–17.8 mg/dL).

No significant differences were detected between ASD-LL and ASD-HL with regard to the time of urine collection (before or after 12h noon; $\chi^2(2) = .373$, $p = .830$), tantrums during the past 24 hours before the urine collection ($\chi^2(2) = .659$, $p = .719$), episodes of fever over the past two weeks ($\chi^2(2) = 5.632$, $p = .060$) and vaccinations ($\chi^2(2) = .002$, $p = .999$), and none of the children were taking metabolic related medication, which are aspects that could influence lactate concentrations in urine [13]. Additional analyses on 90% of the data were conducted to control for the effect of urinary dilution depending upon the amount of liquid children have drunk by means of measurement of creatinine ($M = 81.9$ mg/dL, $SD = 46.9$ mg/dL; range: 7.53–224.47 mg/dL). Creatinine in urine was measured using the CREJ2 (Creatinine Jaffé Gen.2) protocol from Cobas ⓡ Roche diagnostics GmbH as described by the manufacturer. Analyses of the differences in ASD characteristics and regression in children in the group with the lowest and highest lactate/creatinine ratio and analyses of the lactate/creatinine ratio in urine between regression groups showed comparable results with the analyses based on the lactate concentration (see also S4 File for an overview of these additional analyses).

As controls, data of 38 randomly selected children aged between 2 and 11 years old ($M = 6.64$ years, $SD = 2.65$; 39.5% boys) were collected. None of the control children were diagnosed with ASD, confirmed by scores below the ASD range on the Social Communication Questionnaire (SCQ [58]) for all children. Furthermore, none of the parents reported any clinical characteristics related to mitochondrial dysfunction. Analyses showed that the mean concentration of urinary lactate in control children was 5.4 mg/dL ($SD = 2.7$ mg/dL; range: 1.2–13.6 mg/dL) and did not significantly differ ($U(137) = 2000.50$, $p = .566$) from the mean lactate value in urine of the children with ASD (i.e., 6.0 mg/dL). In addition, also the mean concentration of creatinine in urine (79.7 mg/dL, SD = 36.1 mg/dL; range: 19.3–162.6 mg/dL) did not significantly differ ($U(128) = 1686.00$, $p = .900$) from the mean creatinine value in children with ASD (i.e., 81.9 mg/dL). Thus, lactate concentrations detected in children with ASD in the present study were similar to lactate concentrations in a control group of children without ASD. See also Fig 1, Panel B for an overview of urinary lactate concentrations in control children.

**Group membership allocation based on developmental regression.** In line with recent previous research on regression [59,60], we decided to conduct the analyses for the broader group of children with regression in language, social, motor and/or adaptive skills before, at, or after the age of 36 months not caused by brain injury or significant life event (e.g., the birth of a sibling, the start of school or a move). This *inclusive regression group (or ASD-R)* consisted of 37 children. The mean onset age of regression was 35.65 months (2.97 years; $SD = 21.70$ months; range: 10–96 months). Additional analyses were conducted on 2 regression groups based on a 36 months age cut-off. The first group consisted of children with regression *before or at 36 months (ASD-R≤36M; n = 24)*. Within ASD-R≤36M, the mean onset age of regression was 21.67 months (i.e., 1.81 years; $SD = 6.64$ months; range: 10–36 months). In all children, regression involved loss of language and/or social or play skills, in most cases combined with loss of other skills such as motor or adaptive skills. The second group consisted of children with regression *after 36 months (ASD-R>36M; n = 13)*. The mean onset age of children within the ASD-R>36M group was 61.46 months (i.e., 5.12 years; $SD = 14.66$ months; range: 42–96 months). In 61.54% (n = 8) of these children regression involved language and/or social skills, in some of them in combination with loss of other skills. In two and three children only loss of motor or loss of adaptive skills was reported, respectively.

Next to the inclusive regression group (ASD-R) also a *non-regression group (or ASD-NR)* of 62 children was defined. This group consisted of (i) children who were reported to display characteristics of ASD and/or non-specific concerns related to ASD already during the first 12

months of life without loss of skills and (ii) children who had a typical development followed by a plateau or stagnation.

**Statistical analyses.** Statistical analyses were conducted with IBM SPSS version 25.0 [61]. Little's MCAR test [62] confirmed that missing data were 'missing completely at random' ($\chi^2(271) = 287{,}440$, $p = .235$). Missing values and outliers [63] in the data were imputed using the Expectation-Maximization technique [64–66]. Criteria for parametric testing were met, except for the lactate concentrations and the ADOS CSS of which the data followed a non-normal distribution. The family's socioeconomic status (SES) was calculated by the Hollingshead's four factor index based on both parents' education level and occupation [67].

First, clinical characteristics of mitochondrial dysfunction were examined in children with ASD by using descriptive statistics. Second, ASD-HL were compared to ASD-LL with regard to clinical characteristics of mitochondrial dysfunction, severity of ASD characteristics and regression. Third, children with regression before and after the age of 36 months were compared to children without regression concerning clinical characteristics associated with mitochondrial dysfunction and lactate concentration in urine. Group differences were examined using Chi square (categorical variables), one-way ANOVA (continuous variables), and non-parametric Kruskall-Wallis H tests. Parametric post hoc comparisons were conducted by Hochberg's GT2 (because of the large variation in sample sizes) and non-parametric by Mann-Whitney U tests. Bonferroni or Bonferroni-Holm correction to control for the inflation of the Type I error rate due to multiple comparisons was considered. However, both procedures cause a substantial reduction of statistical power [68,69]. Considering the low statistical power due to small sample sizes of the groups and the fact that we aim to detect small differences, we decided not to apply this correction since it would reduce the possibility of finding relevant group differences. Effect sizes were calculated using Cohen's d for continuous variables, odds ratio for binary variables within a 2 x 2 contingency table and *W* for binary variables within a 3 x 3 contingency table.

Due to significant differences in age between regression groups as well as between lactate groups, correlations between age and outcome variables were analysed. Significant correlations were not found except for a significant moderate negative correlation between age and ADOS CSS of RRBs ($\rho = -.63$, $p < .01$), and between age and ADOS CSS of social affect ($\rho = -.22$, $p < .05$). However, since ADOS CSS already take into account the age of the children to enhance comparability over different modules, additional correction for age was not conducted. Further, age was significantly but weakly positively correlated with lactate concentration in urine ($\rho = .26$, $p < .05$). Additional evidence on the decision not to "control" for variable age can be found in the critical commentary by Miller and Chapman (2001) on "correcting" or "controlling for" real group differences on a potential covariate [70].

## Results

### Clinical characteristics reminiscent of MDs in ASD

Clinical characteristics in this sample of children with ASD showed a broad range of phenotypic expression reminiscent of MD with multiple organ systems affected. Clinical characteristics that could fit into the diagnosis of MD in the present sample were GI disease, fine and gross motor impairments, ear impairments (including hearing problems and frequent ear infections), respiratory impairments and hypotonia. Other clinical abnormalities were fatigue (muscle weakness and/or exercise intolerance) and cardiovascular diseases. See also Table 1 for an overview.

**Table 1. Clinical characteristics reminiscent of MDs in ASD.**

|  | ASD ($N = 99$) |
|---|---|
| **Hypotonia %** | 21.2 |
| **Respiratory impairments %** | 25.3 |
| **Eye impairments %** | 26.3 |
| **Ear impairments %** | 40.4 |
| **Fine motor impairments %** | 47.5 |
| **Gross motor impairments %** | 44.4 |
| **Cardiovascular disease %** | 7.1 |
| **Kidney impairments %** | 4 |
| **Gastro-Intestinal (GI) disease %** | 51.5 |
| **Liver impairments %** | 2 |
| **Metabolic impairments %** | 2 |
| **Fatigue problems %** | 12.1 |
| **Seizures %** | 4 |

ASD = Autism spectrum disorder.

## Differences between lactate groups on clinical characteristics reminiscent of MDs, ASD characteristics and regression

**Demographic and clinical characteristics of children in lactate groups.** ASD-HL were significantly older ($M = 8.59$ years, $SD = 1.45$) than ASD-LL ($M = 6.75$ years, $SD = 2.08$; $F$ $(1,38) = 10.557$, $p < .01$, $d = 1.05$). As expected, ASD-HL showed a significant higher urine lactate concentration ($M = 11.5$ mg/dL, $SD = 2.6$ mg/dL) as compared to ASD-LL ($M = 2.1$ mg/dL, $SD = .8$ mg/dL; $F(1,38) = 247.372$, $p < .01$, $d = 5.10$). Although ASD-LL showed a below average intelligence profile ($M = 87.21$; $SD = 24.96$) as compared to ASD-ML and ASD-HL, who showed an average intelligence profile ($M = 91.96$; $SD = 18.31$ and $M = 90.75$; $SD = 23.38$, respectively), the difference between ASD-LL and ASD-HL was not significant ($F(1,38) = .215$, $p = .646$, $d = .15$). Further, significant differences were not found in the proportion of children with intellectual disability (ID; i.e. non-verbal IQ $\leq 70$) between ASD-LL (35%) and ASD-HL (25%; $\chi^2(1) = .476$, $p = .490$, $OR = 1.62$). Children with ASD had different comorbidities such as developmental coordination disorder.

(DCD), attention deficit hyperactivity disorder (ADHD), dyspraxia (motor and language), epilepsy, obsessive compulsive disorder (OCD) and language disorders. Within ASD-LL 3 children (15%) and in ASD-HL 5 children (25%) were reported to have an additional diagnosis which related to impairments in motor functioning, however, this difference was not significant ($FET\ p = .695$, $OR = 1.89$). Furthermore, within ASD-LL less children were reported to have an additional diagnosis of ADHD ($n = 2$; 10%) compared to ASD-HL ($n = 8$; 40%) and this difference was marginally significant ($FET\ p = .065$, $OR = 6.00$) showing a statistical significant trend and a large clinical significance ($OR > 5$). See also Table 2.

**Clinical characteristics reminiscent of MDs within children in lactate groups.** Statistical significant differences between the different lactate groups concerning several clinical characteristics suggestive for MD were not found. With regard to clinical significance, a large difference ($OR = 6.33$) was found indicating that ASD-LL, although not statistically significant, showed more fatigue problems as compared to ASD-HL. See also Table 3 for an overview.

**Clinical characteristics reminiscent of MDs within family members of children in lactate groups.** Family members of ASD-HL showed significantly more depressive disorders ($n = 14$; 70%) as compared to ASD-LL ($n = 6$; 30%; $\chi^2(1) = 6.400$, $p < .05$, $OR = 5.44$). Further,

**Table 2. Demographic and clinical differences between children belonging to the different lactate groups.**

| | ASD-LL (n = 20) | ASD-ML (n = 59) | ASD-HL (n = 20) | Differences ASD-LL and ASD-HL | | |
| --- | --- | --- | --- | --- | --- | --- |
| | | | | Statistic (df) | p | ES [a] |
| **Age in years** *M* (*SD*) | 6.75 (2.08) | 7.47 (1.99) | 8.59 (1.45) | 10.557 (1,38) [b] | .002* | 1.05 |
| **Male sex** *F* (%) | 14 (70) | 48 (81.36) | 10 (50) | 1.667 (1) [c] | .197 | 2.33 |
| **SES** *M* (*SD*) | 42.85 (11.07) | 43.05 (12.96) | 40.38 (14.91) | .355 (1,38) [b] | .555 | .19 |
| **Lactate mg/dL** *M* (SD) | 2.1 (.8) | 5.4 (1.8) | 11.5 (2.6) | 247.372 (1,38) [b] | .000** | 5.10 |
| **Lactate mg/dL** *Range* | .5–3.2 | 3.2–8.9 | 9.0–17.8 | / | / | / |
| **Non-Verbal IQ** *M* (*SD*) | 87.21 (24.96) | 91.96 (18.31) | 90.75 (23.38) | .215 (1,38) [b] | .646 | .15 |
| **ID (IQ≤70)** *F (%)* | 7 (35) | 8 (13.56) | 5 (25) | .476 (1) [c] | .490 | 1.62 |
| **Comorbidities** *F (%)* | 8 (40) | 14 (23.73) | 9 (45) | .102 (1) [c] | .749 | 1.23 |
| **DCD or dyspraxia** *F (%)* | 3 (15) | 7 (11.86) | 5 (25) | / [d] | .695 | 1.89 |
| **ADHD** *F (%)* | 2 (10) | 7 (11.86) | 8 (40) | / [d] | .065 | 6.00 |
| **Epilepsy** *F (%)* | 2 (10) | 2 (3.39) | 0 (0) | / [d] | .487 | / |
| **Special education** *F/n(%)* | 10/19 (52.63) | 21/57 (36.84) | 8/19 (42.11) | .422 (1) [c] | .516 | 1.53 |
| **Therapy** *F/n (%)* | 16/19 (84.21) | 51/56 (91.07) | 16/19 (84.21) | / [d] | 1.000 | 1.00 |

ASD-LL = children within the low lactate group, ASD-ML = children within the medium lactate group, ASD-HL = children within the high lactate group, SES = Social Economic Status; ID = Intellectual Disability; DCD = Developmental Coordination Disorder; ADHD = Attention Deficit Hyperactivity Disorder

[a] ES effect size = Cohen's *d* for continuous variables, odds ratio for binary variables within a 2 x 2 contingency table

[b] F test

[c] Chi square test

[d] Fisher's Exact Test

* *p* < .01

** *p* < .001.

**Table 3. Clinical characteristics typically seen in MDs within children belonging to the different lactate groups.**

| | ASD-LL (n = 20) | ASD-ML (n = 59) | ASD-HL (n = 20) | Differences ASD-LL and ASD-HL | | |
| --- | --- | --- | --- | --- | --- | --- |
| | | | | Statistic (df) | p | ES [a] |
| **Hypotonia** *F (%)* | 6 (30) | 9 (15.25) | 6 (30) | .000 (1) [b] | 1.000 | 1.00 |
| **Respiratory impairments** *F (%)* | 8 (40) | 12 (20.34) | 5 (25) | 1.026 (1) [b] | .311 | 2.00 |
| **Eye impairments** *F (%)* | 3 (15) | 16 (27.12) | 7 (35) | / [c] | .273 | 3.05 |
| **Ear impairments** *F (%)* | 10 (50) | 25 (42.37) | 5 (25) | 2.667 (1) [b] | .102 | 3.00 |
| **Cardiovascular disease** *F (%)* | 1 (5) | 6 (10.17) | 0 (0) | / [c] | 1.000 | / |
| **Kidney impairments** *F (%)* | 1 (5) | 1 (1.69) | 2 (10) | / [c] | 1.000 | 2.11 |
| **Gastro-Intestinal disease** *F (%)* | 10 (50) | 32 (54.24) | 9 (45) | .100 (1) [b] | .752 | 1.22 |
| **Liver impairments** *F (%)* | 2 (10) | 0 (0) | 0 (0) | / [c] | .487 | / |
| **Metabolic impairments** *F (%)* | 2 (10) | 0 (0) | 0 (0) | / [c] | .487 | / |
| **Fatigue problems** *F (%)* | 5 (25) | 6 (10.17) | 1 (5) | / [c] | .182 | 6.33 |

ASD-LL = children within the low lactate group, ASD-ML = children within the medium lactate group, ASD-HL = children within the high lactate group

[a] ES effect size = odds ratio for binary variables within a 2 x 2 contingency table

[b] Chi square test

[c] Fisher's Exact Test.

**Table 4. Clinical characteristics reminiscent of MDs within family members of children belonging to the different lactate groups.**

| | ASD-LL (n = 20) | ASD-ML (n = 59) | ASD-HL (n = 20) | Differences ASD-LL and ASD-HL | | |
|---|---|---|---|---|---|---|
| | | | | *Statistic* (df) | *p* | ES [a] |
| **Intellectual disability** *F (%)* | 3 (15) | 16 (27.12) | 6 (30) | / [c] | .451 | 2.43 |
| **Epilepsy** *F (%)* | 2 (10) | 1 (1.69) | 4 (20) | / [c] | .661 | 2.25 |
| **ADHD or ADD** *F (%)* | 7 (35) | 3 (5.08) | 8 (40) | .107 (1) [b] | .744 | 1.24 |
| **Anxiety disorder** *F (%)* | 4 (20) | 11 (18.64) | 6 (30) | / [c] | .716 | 1.71 |
| **Depressive disorder** *F (%)* | 6 (30) | 24 (40.68) | 14 (70) | 6.400 (1) [b] | .011* | 5.44 |
| **Immune disorder** *F (%)* | 1 (5) | 14 (23.73) | 7 (35) | / [c] | .044* | 10.23 |
| **Mitochondrial disorder** *F (%)* | 0 (0) | 2 (3.39) | 1 (5) | / [c] | 1.000 | / |
| **Diabetes type I** *F (%)* | 1 (5) | 9 (15.25) | 2 (10) | / [c] | 1.000 | 2.11 |
| **Cardiovascular disease** *F (%)* | 5 (25) | 23 (38.98) | 4 (20) | / [c] | 1.000 | 1.33 |
| **Muscular disease** *F (%)* | 0 (0) | 5 (8.47) | 4 (20) | / [c] | .106 | / |
| **Liver disease** *F (%)* | 0 (0) | 6 (10.17) | 2 (10) | / [c] | .487 | / |
| **Chronic Fatigue Syndrome** *F (%)* | 1 (5) | 10 (16.95) | 2 (10) | / [c] | 1.000 | 2.11 |
| **Fibromyalgia** *F (%)* | 2 (10) | 4 (6.77) | 2 (10) | / [c] | 1.000 | 1.00 |
| **Multiple Sclerosis** *F (%)* | 1 (5) | 1 (1.69) | 0 (0) | / [c] | 1.000 | / |
| **Parkinson** *F (%)* | 1 (5) | 1 (1.69) | 0 (0) | / [c] | 1.000 | / |

ASD-LL = children within the low lactate group, ASD-ML = children within the medium lactate group, ASD-HL = children within the high lactate group,

ADHD = Attention Deficit Hyperactivity Disorder, ADD = Attention Deficit Disorder

[a] ES effect size = Cohen's *d* for continuous variables

[b] Chi square test

[c] Fisher's Exact Test

* *p* < .05.

family members of ASD-HL were also reported to show more disorders related to the immune system (*n* = 7; 35%) than family members of ASD-LL (*n* = 1; 5%; *FET p* < .05, *OR* = 10.23). See also Table 4 for an overview of group differences on several clinical characteristics of MD within the family. With regard to genetic disorders, 2 family members of ASD-HL were reported to have Angelman syndrome and Prader-Willi syndrome respectively. Also 6 individuals with a chromosome disorder were detected within the families of ASD-LL and ASD-ML. Parents reported no other genetic disorders related to ASD or regression. Two individuals within the families of ASD-ML were reported to have Alzheimer disease.

**Characteristics of ASD and regression in children in lactate groups.** Significant differences were not found in the ADOS-2 CSS and prevalence of children with and without regression between the low and high lactate groups. See also Table 5.

## Differences between regression groups on clinical characteristics typically seen in MDs and lactate concentration in urine

**Demographic and clinical characteristics of children in regression groups.** ASD-NR were significantly older (*M* = 8.09 years, *SD* = 1.85) than the inclusive group ASD-R (*M* = 6.65 years, *SD* = 1.90; *F*(1,97) = 13.906, *p* < .01, *d* = .78). Significant differences were detected in chronological age (*F*(2,96) = 8.218, *p* < .01, *d* = .72) between ASD-NR, ASD-R≤36M and ASD-R>36M. Post hoc comparisons revealed that ASD-NR were significantly older (*p* = .000) than ASD-R≤36M (*M* = 6.30 years, *SD* = 2.07). No significant differences were found between ASD-NR and ASD-R>36M (*p* = .392) and, ASD-R≤36M and ASD-R>36M (*p* = .339).

**Table 5. Differences in ASD characteristics and regression in the children belonging to the different lactate groups.**

| | ASD-LL (n = 20) | ASD-ML (n = 59) | ASD-HL (n = 20) | Differences ASD-LL and ASD-HL Statistic (df) | p | ES [a] |
|---|---|---|---|---|---|---|
| **ADOS-2 Total CSS** *M (SD)* | 6.10 (2.77) | 6.32 (2.27) | 6.80 (2.33) | 226 (40) [b] | .495 | .22 |
| *Mean rank* | 19.20 | | 21.80 | | | |
| **ADOS-2 Soc Affect CSS** *M (SD)* | 5.65 (2.25) | 6.15 (2.26) | 6.80 (2.22) | 252.50 (40) [b] | .157 | .46 |
| *Mean rank* | 17.88 | | 23.12 | | | |
| **ADOS-2 RRB CSS** *M (SD)* | 8.10 (2.51) | 7.19 (2.29) | 6.95 (2.69) | 142.50 (40) [b] | .121 | .51 |
| *Mean rank* | 23.38 | | 17.62 | | | |
| **ASD-NR** *F (%)* | 10 (50) | 38 (64.41) | 14 (70) | 1.667 (1) [d] | .197 | 2.33 |
| **ASD-R≤36M** *F (%)* | 7 (35) | 14 (23.73) | 3 (15) | / [e] | .273 | 3.05 |
| **ASD-R > 36M** *F (%)* | 3 (15) | 7 (11.86) | 3 (15) | / [e] | 1.000 | 1.00 |

ASD-LL = children within the low lactate group, ASD-ML = children within the medium lactate group, ASD-HL = children within the high lactate group, ADOS-2 = Autism Diagnostic Observation Scale Second Edition, CSS = Calibrated Severity Score, ASD-R≤36M = children with regression before or at 36 months, ASD-R>36M = children with regression after 36 months

[a] ES effect size = Cohen's *d* for continuous variables and odds ratio for binary variables within a 2 x 2 contingency table

[b] Mann Whitney U Test

[c] F test

[d] Chi square test

[e] Fisher's exact test.

Further, no significant differences were found in gender or SES between ASD-NR and regression groups. ASD-NR showed an average intelligence profile in contrast to the regression groups which showed a below average intelligence profile. However, the difference between ASD-NR and regression groups was not significant. Also, no significant differences were found in the proportion of children with ID between ASD-NR and regression groups. With regard to comorbidities in general, ASD-NR showed significantly more comorbidities (*n* = 24; 38.71%) as compared to ASD-R (*n* = 7; 18.92%; $\chi^2(1)$ = 4.220, *p* < .05, *OR* = 2.71). More precisely, ASD-R ≤36M showed significantly less comorbidities (*n* = 3; 12.50%) than ASD-NR (*FET p* < .05). Further, significantly more ASD-NR were reported to have a comorbid diagnosis of DCD or dyspraxia (*n* = 13; 20.97%) as compared to ASD-R (*n* = 2; 5.41%; *FET p* < .05, *OR* = 4.64). Again, ASD-R ≤36M showed significantly less DCD or dyspraxia (*n* = 0) than ASD-NR (*FET p* < .05). A significant difference between ASD-NR and regression groups with regard to the diagnosis of ADHD or the presence of epilepsy was not found. Further, only a large clinical significant difference (*OR* = 6.6) was found indicating that more ASD-R (97.06%) followed therapy related to ASD characteristics compared to ASD-NR (83.33%). See also 3.068 (2)Table 6.

**Clinical characteristics reminiscent of MDs within children in regression groups.** ASD-NR showed significantly more cardiovascular diseases as compared to ASD-R (*FET p* < .05). Although no significant difference was found with regard to respiratory impairments between ASD-NR and the regression groups, post hoc tests revealed that ASD-R>36M showed significantly more respiratory impairments compared to ASD-R≤36M (*FET p* < .05). No other significant differences were found between regression groups on several clinical characteristics of MD. See also Table 7 for an overview.

**Lactate in urine in regression groups.** No significant difference was found between the average lactate concentrations in ASD-NR (*M* = 6.3 mg/dL, *SD* = 3.9 mg/dL; mean rank = 52.3

**Table 6. Demographic and clinical differences between children belonging to different regression groups.**

| | ASD-NR (n = 62) | ASD-R (n = 37) | ASD-R ≤36M (n = 24) | ASD-R >36M (n = 13) | Differences ASD-NR and ASD-R | | | Differences ASD-NR, ASD-R≤36M and ASD-R>36M | | | |
| --- | --- | --- | --- | --- | --- | --- | --- | --- | --- | --- | --- |
| | | | | | Statistic (df) | p | ES [a] | Statistic (df) | p | ES [a] | Post hoc |
| **Age in years** M (SD) | 8.09 (1.85) | 6.65 (1.90) | 6.30 (2.07) | 7.28 (1.37) | 13.906 (1,97) [b] | .000*** | .78 | 8.218 (2,96) [b] | .001** | .72 | ASD-NR > ASD≤36M |
| **Male sex** F (%) | 48 (77.42) | 24 (64.86) | 17 (70.83) | 7 (58.86) | 1.841 (1) [c] | .175 | 1.86 | 3.068 (2) [c] | .216 | .07 | / |
| **SES** M (SD) | 41.65 (12.80) | 43.84 (13.23) | 43.13 (14.64) | 45.15 (10.55) | .658 (1,97) [b] | .419 | .17 | .429 (2,96) [b] | .652 | .15 | / |
| **Non-Verbal IQ** M (SD) | 93.67 (19.38) | 85.88 (22.18) | 84.64 (22.58) | 88.15 (22.13) | 3.358 (1,97) [b] | .070 | .39 | 1.789 (2,96) [b] | .173 | .35 | / |
| **ID (IQ≤70)** F (%) | 9 (14.52) | 11 (29.73) | 8 (33.33) | 3 (23.08) | 3.327 (1) [c] | .068 | 2.49 | 3.877 (2) [c] | .144 | .20 | / |
| **Comorbidities** F/n (%) | 24/62 (38.71) | 7/37 (18.92) | 3/24 (12.50) | 4/13 (30.77) | 4.220 (1) [c] | .040* | 2.71 | 5.528 (2) [c] | .063 | .24 | ASD-NR > ASD-R≤36M |
| **DCD or dyspraxia** F (%) | 13 (20.97) | 2 (5.41) | 0 (0) | 2 (15.38) | / | .044* | 4.64 | 5.918 (2) [c] | .052 | .24 | ASD-NR > ASD-R≤36M |
| **ADHD** F (%) | 13 (20.97) | 4 (10.81) | 1 (4.17) | 3 (23.08) | / [d] | .273 | 2.19 | 3.801 (2) [c] | .150 | .20 | / |
| **Epilepsy** F (%) | 2 (3.23) | 2 (5.41) | 1 (4.17) | 1 (7.69) | / [d] | .594 | 1.71 | .554 (2) [c] | .758 | .07 | / |
| **Special education** F/n(%) | 25/61 (40.98) | 14/34 (41.18) | 11/23 (47.83) | 3/11 (27.27) | .000 (1) [c] | .985 | .99 | 1.299 (2) [c] | .522 | .11 | / |
| **Therapy** F/n (%) | 50/60 (83.33) | 33/34 (97.06) | 23/23 (100) | 10/11 (90.90) | / [d] | .053 | 6.6 | 4.552 (2) [c] | .103 | .21 | / |

ASD-NR = children without regression, ASD-R = children who ever experienced a regression, ASD-R≤36M = children with regression before or at 36 months,

ASD-R>36M = children with regression after 36 months, SES = Social Economic Status, ID = Intellectual Disability, DCD = Developmental Coordination Disorder,

ADHD = Attention Deficit Hyperactivity Disorder

[a] ES effect size = Cohen's d for continuous variables, odds ratio for binary variables within a 2 x 2 contingency table and W for binary variables within a 3 x 3

contingency table

[b] F test

[c] Chi square test

[d] Fisher's Exact Test

* p < .05

** p < .01

*** p < .001.

mg/dL) and ASD-R (M = 5.4 mg/dL, SD = 3.00 mg/dL; mean rank = 46.2 mg/dL; U(99) = 1,004.5, p = .303, d = .21). With regard to the 36 months age cut-off, also no significant difference was found between ASD-NR, ASD-R≤36M (M = 5.4 mg/dL, SD = 2.8 mg/dL; mean rank = 46.1 mg/dL) and ASD>36M (M = 5.4 mg/dL, SD = 3.4 mg/dL; mean rank = 46.3 mg/dL; H(2) = 1.063, p = .588, d = .20).

## Discussion

Although our cohort of children with ASD showed a broad range of phenotypic characteristics that could fit into the diagnosis of MD, the results of the urinary lactate concentrations did not provide an additional argument in favor for an underlying mitochondrial defect. The mean urinary lactate value in children with ASD did not significantly differ from control children. Furthermore, lactate concentration was checked in blood in the two ASD participants with the highest lactate concentrations in urine, and this turned out to be normal. Also, no significant differences were found between individuals with low and high lactate concentrations in urine

**Table 7. Clinical characteristics reminiscent of MDs within children belonging to different regression groups.**

| | ASD-NR (n = 62) | ASD-R (n = 37) | ASD-R ≤36M (n = 24) | ASD-R>36M (n = 13) | Differences ASD-NR and ASD-R | | | Differences ASD-NR, ASD-R≤36M and ASD-R>36M | | | |
|---|---|---|---|---|---|---|---|---|---|---|---|
| | | | | | Statistic (df) | p | ES [a] | $X^2$ (2) | p | ES [a] | Post hoc |
| Hypotonia F (%) | 11 (17.74) | 10 (27.03) | 7 (29.17) | 3 (23.08) | 1.195 (1) [b] | .274 | .23 | 1.382 | .501 | .12 | / |
| Respiratory impairm. F (%) | 16 (25.81) | 9 (24.32) | 3 (12.5) | 6 (46.15) | .027 (1) [b] | .870 | .03 | 5.087 | .079 | .23 | ASD-R≤36M < ASD-R>36M |
| Eye impairments F (%) | 17 (27.42) | 9 (24.32) | 4 (16.67) | 5 (20.83) | .115 (1) [b] | .735 | .07 | 2.183 | .336 | .15 | / |
| Ear impairments F (%) | 24 (38.71) | 16 (43.24) | 9 (37.5) | 7 (53.85) | .198 (1) [b] | .657 | .09 | 1.133 | .567 | .11 | / |
| Cardiovascular disease F (%) | 7 (11.29) | 0 (0) | 0 (0) | 0 (0) | / [c] | .043* | / | 4.495 | .106 | .21 | / |
| Kidney impairments F (%) | 4 (6.45) | 0 (0) | 0 (0) | 0 (0) | / [c] | .294 | / | 2.488 | .288 | .16 | / |
| Gastro-Intestinal disease F(%) | 34 (54.84) | 17 (45.95) | 11 (45.83) | 6 (46.15) | .734 (1) [b] | .392 | .18 | .734 | .693 | .09 | / |
| Liver impairments F (%) | 1 (1.62) | 1 (2.70) | 1 (4.17) | 0 (0) | / [c] | 1.000 | 1.00 | .879 | .644 | .09 | / |
| Metabolic impairments F (%) | 1 (1.62) | 1 (2.70) | 0 (0) | 1 (7.69) | / [c] | 1.000 | 1.00 | 2.660 | .265 | .16 | / |
| Fatigue problems F (%) | 7 (11.29) | 5 (13.51) | 2 (8.33) | 3 (12.5) | .108 (1) [b] | .743 | .07 | 1.828 | .401 | .14 | / |

ASD-NR = children without regression, ASD-R = children who ever experienced a regression, ASD-R≤36M = children with regression before or at 36 months,
ASD-R>36M = children with regression after 36 months

[a] ES effect size = Cohen's $d$ for continuous variables, odds ratio for binary variables within a 2 x 2 contingency table and $W$ for binary variables within a 3 x 3 contingency table

[b] F test

[c] Fisher's Exact Test

* $p < .05$.

with regard to several clinical characteristics of mitochondrial dysfunctions such as respiratory, hearing and vision impairments. Overall, the results in the present study are in line with one of the most comprehensive studies to date examining the role of mitochondrial variation in nearly 1,300 individuals with ASD and 2,600 controls [71]. After sequencing or genotyping mtDNA, no evidence was found to suggest a major role for mtDNA variation in ASD susceptibility, indicating that "mtDNA variation is not a major contributing factor to the development of ASD" [71].

With regard to the prevalence of reported regression, no differences were found between the group of ASD children with low concentrations of lactate in urine and the group of ASD children with high concentrations of lactate in urine. Moreover, participants with ASD and regression (before or after 36 months) did not present a significantly higher concentration of lactate in urine as compared to ASD children without regression. These findings are in contrast with recent preliminary evidence for a unique type of mitochondrial dysfunction (i.e., abnormal respiratory function characterised by increased maximal oxygen consumption rate, maximal respiratory capacity and reserve capacity) which may be related to regression in a small clinical sample of children with ASD [72]. In addition to more prevalent regression, Rossignol and Frye (2012) also demonstrated a greater prevalence of seizures, motor delay and GI

abnormalities in ASD children with MD, or with abnormal biochemical markers for mitochondrial function [11]. In the present study, however, no differences between lactate groups were found concerning epilepsy, comorbidities related to impaired motor function and GI abnormalities.

Regarding the link between severity of core ASD characteristics and increased lactate concentration in urine, significant differences between low and high lactate groups were not detected. These results are in contradiction with two small-scale studies [73,74] that reported significant associations between mitochondrial dysfunctions and severity of ASD. The first study reported a significant correlation between abnormal levels of brain markers of mitochondrial function and severity of language, and neuropsychological deficits only in the ASD group ($n$ = 11) as compared to a typically developing group of children ($n$ = 11) [73]. In another study, higher concentrations of lactate and lower levels of carnitine were detected in blood of 30 children between 4 and 12 years of age with severe ASD characteristics as compared to individuals with mild or moderate ASD characteristics [74].

Of interest is that both depressive and immune disorders were found to be more common in families of children with higher lactate concentrations in urine. Further, with regard to genetic disorders, two family members of children with high lactate concentrations were reported to have Angelman syndrome and Prader-Willi syndrome (PWS), respectively. Recently, preliminary results in six children and adults with PWS, suggested a decreased mitochondrial function as compared to healthy controls with significant differences in basal respiration, maximal respiratory capacity and ATP-linked respiration [75]. Furthermore, high lactate concentration was associated with older chronological age and more ADHD comorbidity. The association with ADHD is explained by some authors based on the fact that lactate concentration can be increased after physical activity [45,76,77]. Furthermore, since the brain is the most energetic tissue in the body and thus highly reliant on mitochondrial energy, partial systemic mitochondrial dysfunctions can predispose not only to ASD but also to other neuropsychiatric disorders such as ADHD and depressive disorders [26,46].

## Strengths and limitations

Next to the large sample size of this exploratory study, there are several other strengths. First, most of the previous studies reporting abnormal biochemical markers of mitochondrial function in ASD used a clinical referred sample. Therefore, in the present study a community-based sample which is less susceptible to referral bias was included [11]. Furthermore, in the present study children with ASD with a wide range of cognitive ability were included whereas in some of the prior studies [78] all ASD children presented with developmental delay and/or cognitive deficits which can result from mitochondrial dysfunction and thus evoke higher prevalence of MD in ASD. Second, although increases in lactate concentrations in blood have been consistently reported as a potential biomarker for abnormal mitochondrial metabolism in children with ASD [11,35] it is important to notice that this is an invasive technique prone to vary between samplings, for instance excessive muscle movement could lead to falsely increased lactate concentrations, especially in young children who are struggling or because the tourniquet is left on too long [25,30]. Therefore, the present study aimed to measure lactate in urine as this non-invasive technique could be an alternative method to screen for a metabolic defect, in particular a mitochondrial defect, in children with ASD.

We need to acknowledge that the present study has several limitations. Most importantly, we cannot rule out that an association between metabolic abnormalities and regression in ASD could have been missed by choice of the biomarker, i.e. lactate concentration in urine. In a previous study [79], it was shown that urinary lactate concentrations correlate less well with

various known MDs as compared to blood lactate. Only 27% of the patients with MD had lactate values in urine above the upper limit of normal, and 13% of samples from unselected patients also had values above the normal range. Thus, urinary lactate concentration was not increased in the majority of the patients with MD although most of them had blood lactate concentrations above the diagnostic cut-off for presence of mitochondrial dysfunction [79]. In addition, the patients with MD, especially those with milder phenotype, can present with normal lactate concentration in both urine and blood [25]. In the present study, the clinical characteristics in the sample of children with ASD showed the expected broad range of clinical characteristics as seen in MD with multiple systems affected [19,28]) Therefore (milder) mitochondrial dysfunction cannot be excluded for the children in the low (and medium) lactate group. Another limitation of the present study is the retrospective parent report of developmental regression. Although efficient and cost-effective, the parent report method has several limitations concerning validity including recall problems [80,81]. To tackle the limitations of retrospective methods, studies into developmental pathways in ASD could benefit from the use of a prospective research approach in which the development of infants and toddlers who are at elevated likelihood for ASD (EL-sibs [82]) is followed-up until the age of 36 months at which levels of ASD characteristics are assessed and for most children a diagnosis can be reliably confirmed or ruled out [83]. Different prospective studies demonstrated that more gradual longitudinal declines in subtle social-communicative behaviours between the ages of 6 to 36 months were present in the majority of children who develop ASD [84–86]. If mitochondrial dysfunctions are related to skill loss, findings on the presence of subtle declines in social-communicative development in the majority of children with ASD could indicate that mild mitochondrial dysfunctions are a common characteristic of ASD. This is also in line with the suggestion of Rossignol and Frye (2012) concerning the presence of a 'spectrum' of mitochondrial dysfunction in children with ASD [11]. On the other hand, it is also possible that the type of regression resulting from MD differs from the mild declines reported in the prospective studies. The regression in MD is more dramatic, with rapid loss of skills. This was less the case in the prospective studies, but reported by parents in a substantial subset of children with ASD in retrospective studies [87]. However, at the moment there is no overall consensus with regard to the definition, operationalization and aetiology of regression in ASD and integration of retrospective and prospective findings is required to gain more insight into the nature of regression in ASD [81].

## Implications

Despite increasing evidence on the association between mitochondrial dysfunction and ASD, the present outcomes are in line with outcomes from a previous comprehensive genetic study indicating that it is not a major contributing factor to the development of ASD [71]. Hence, we can conclude that the evidence for this challenging association is still preliminary to date, also given that most conclusions are based on relatively small sample sizes and great variety in screening methods. Therefore, future research on mitochondrial dysfunction in ASD should focus on using large samples of infants and children with ASD, implementing serial measurements of different biomarkers. As lactate concentrations in urine fluctuate, serial measurements during several days would also be preferred above one single measurement. As an alternative for lactate in plasma and urine, other biochemical markers including caspase 7, glutathione and glutathione S-Transferase, which showed abnormal high concentrations in children with ASD, could discriminate between MD and typically developing children [35]. Furthermore, future studies could also include non-invasive, new imaging techniques such as Magnetic Resonance Spectroscopy (MRS [88]) and Near-Infrared Spectroscopy (NIRS [42]). It

has been established that NIRS allows detection of mitochondrial dysfunction in patients with MD [89]. Furthermore, prospective longitudinal studies investigating the development of infants who are at elevated likelihood for ASD such as the younger siblings of children with ASD [82,90] could implement measurements of regression as well as different biomarkers and imaging techniques to screen for mitochondrial dysfunction at several moments early in life to reveal the association between MD, regression and ASD characteristics. In addition, these studies could also provide more insight into what mitochondrial dysfunctions could look like in individuals with ASD which could be helpful in defining and understanding the clinical characteristics.

Given that the role of mitochondrial dysfunctions in ASD is currently unclear, some investigators have suggested that mitochondrial dysfunction should only be considered during evaluation when typical neurological findings associated with mitochondrial dysfunction are found, or the family history is positive for MD [91–93]. In this respect, it would be premature to handle children with ASD as having a mitochondrial dysfunction only based on increased biomarkers. Additionally, it has not been clearly established yet how mitochondrial bioenergetic deficiencies might be involved in the multifactorial pathogenesis of ASD but one hint lies in the link between mitochondrial dysfunction and cellular oxidative stress [78].

On the other hand, if mitochondrial dysfunctions would be involved in the development of ASD, early identification of children with underlying mitochondrial dysfunction who might be at risk of undergoing regression into ASD, because of a metabolic stressor, could lead to prophylactic measures to prevent the development of ASD, or at least could decrease the severity of core ASD characteristics [11]. Especially, in case an individual with ASD shows recurrent episodes of regression in combination with multiple tissue or organ dysfunctions an extended evaluation for MD is indicated [13,14,94].

Although uncovering the precise genetic and environmental factors that contribute to neuropsychiatric disorders such as ASD is made difficult by the complexity of mitochondrial genetics, it is possible that all of the complex genetic interactions may share mitochondrial bioenergetics deficiency as a common pathophysiological effect and, therefore, be treatable by bioenergetics interventions [26] or metabolism-based therapies such as the high fat, low-carbohydrate ketogenic diet [95].

## Conclusions

In conclusion, an association of increased urinary lactate concentrations as screening method for metabolic abnormalities and possible mitochondrial dysfunction and regression could not be demonstrated in our large ASD cohort of children between 3 and 11 years old. However, since children with ASD showed a broad range of phenotypic characteristics reminiscent of MD such as gastro-intestinal problems, auditory dysfunction, respiratory impairments and hypotonia, the presence of milder mitochondrial dysfunction cannot be excluded. Despite increasing evidence on the association between mitochondrial dysfunction and ASD, the present outcomes are in line with outcomes from a previous comprehensive genetic study indicating that it is not a major contributing factor to the development of ASD. Because of important theoretical and clinical implications, additional research is needed concerning the link between mitochondrial dysfunction and regression in ASD. It is essential that future studies focus more on defining and understanding the clinical characteristics of individuals with ASD and concomitant mitochondrial dysfunctions. Therefore, alternative and more reliable biomarkers and new non- or minimally invasive techniques have to be developed for diagnosing MD in general as well as identifying children with ASD+MD. These could play a major role in unravelling the link between ASD and MD and improve early detection of ASD.

## Supporting information

**S1 File. Recruitment and research flow chart.**
(DOCX)

**S2 File. Interview anamnesis metabolic disorders in the child.**
(DOCX)

**S3 File. Interview medical and psychiatric history of the child and family.**
(DOCX)

**S4 File. Additional analyses lactate/creatinine ratio concentration in urine.**
(DOCX)

## Acknowledgments

We want to thank Cerba Healthcare Belgium–CRI (Zwijnaarde, East Flanders, Belgium) for medical analysis of the urine samples in the study. Further, we are very grateful to the parents and children who participated in the study.

## Author Contributions

**Conceptualization:** Sofie Boterberg, Elise Vantroys, Rudy Van Coster, Herbert Roeyers.

**Data curation:** Sofie Boterberg.

**Formal analysis:** Sofie Boterberg.

**Funding acquisition:** Sofie Boterberg, Rudy Van Coster, Herbert Roeyers.

**Investigation:** Sofie Boterberg, Elise Vantroys.

**Methodology:** Sofie Boterberg, Elise Vantroys, Rudy Van Coster, Herbert Roeyers.

**Project administration:** Sofie Boterberg.

**Resources:** Rudy Van Coster, Herbert Roeyers.

**Supervision:** Rudy Van Coster, Herbert Roeyers.

**Visualization:** Sofie Boterberg.

**Writing – original draft:** Sofie Boterberg, Elise Vantroys.

**Writing – review & editing:** Boel De Paepe, Rudy Van Coster, Herbert Roeyers.

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
