## [Decision Letter · Decision Letter 0]

6 Jul 2022

PONE-D-21-39785Urine lactate concentration as a non-invasive screener for metabolic changes: Findings in children with autism spectrum disorder and regression.PLOS ONE

Dear Dr. Boterberg,

Thank you for submitting your manuscript to PLOS ONE. After careful consideration, we feel that it has merit but does not fully meet PLOS ONE’s publication criteria as it currently stands. Therefore, we invite you to submit a revised version of the manuscript that addresses the points raised during the review process.

We look forward to receiving your revised manuscript.

Kind regards,

Wen-Jun Tu

Academic Editor

PLOS ONE

Journal Requirements:

“ SB is supported by the Research Foundation – Flanders; Belgium (FWO – 12Z8821N/ URL: https://www.fwo.be/). EV is supported by the Special Research Fund Belgium – Interdisciplinary Research Project (grant number: BOF.24J.2014.0005.02/ URL: https://www.ugent.be/en/research/funding/bof/iop). The funders had no role in study design, data collection and analysis, decision to publish, or preparation of the manuscript.”

Please provide an amended statement that declares *all* the funding or sources of support (whether external or internal to your organization) received during this study, as detailed online in our guide for authors at http://journals.plos.org/plosone/s/submit-now.

 Please also include the statement “There was no additional external funding received for this study.” in your updated Funding Statement.

Additional Editor Comments (if provided):

1. What has previously been published on this topic and what does this work add to the existing literature?

2.Is this a prospective study or a retrospective design? Research flow chart needs to be provided. How many patients included and excluded, why? How many patients lost in the follow-up or died in the discharge?

3. This study had been approved by the Ethics Committee？The approval no. of Ethics Committee should be listed and the informed consent was written and/ or oral?

4.When and how the Urine samples were collected? Only one-time point was collected? How do Urine lactate concentration change over the course of follow-up or in the hospital?

Reviewers' comments:

Reviewer's Responses to Questions

**Comments to the Author**

1. Is the manuscript technically sound, and do the data support the conclusions?

Reviewer #1: Yes

2. Has the statistical analysis been performed appropriately and rigorously? 

Reviewer #1: Yes

3. Have the authors made all data underlying the findings in their manuscript fully available?

Reviewer #1: Yes

4. Is the manuscript presented in an intelligible fashion and written in standard English?

Reviewer #1: Yes

5. Review Comments to the Author

Reviewer #1: Interesting research article describing null association between urinary lactate concentrations and autism.

Discussion and limitations sections are informative even for describing negative results.

Statistical analysis has been well performed.

6. PLOS authors have the option to publish the peer review history of their article (what does this mean?). If published, this will include your full peer review and any attached files.

Reviewer #1: No

---

## [Author Response · Author response to Decision Letter 0]

22 Jul 2022

Response to Reviewers

General comments

In general we want to thank the academic editor and reviewer for their important and thoughtful comments which played a major role in the revision of the manuscript. As a major addition to the article, we now adapted the title (Urine lactate concentration as a non-invasive screener for metabolic abnormalities: Findings in children with autism spectrum disorder and regression) since the previous terminology ‘metabolic change’ could lead to interpreting the cross-sectional (retrospective) nature of the study as a prospective longitudinal design. We prefer the term ‘abnormalities’ and hope that the purpose of the study and applied design is now more explicit.

According to the other comments, we also adapted the Introduction-, Method- and Discussion-section of the current study (see below for details). We sincerely hope that the background and contribution of the present study results and, recruitment and methodological aspects are now more clear.

The text that has been adapted and referenced in this cover letter (in red) is also indicated with Track Changes in the revised manuscript (version with Track Changes). 

Journal Requirements:

Requirement # 1: Please ensure that your manuscript meets PLOS ONE's style requirements, including those for file naming. 

Author response # 1: We have consulted the information on the formatting details and adapted the revised manuscript and additional files accordingly.

Requirement # 2: Thank you for stating in your Funding Statement:

“ SB is supported by the Research Foundation – Flanders; Belgium (FWO – 12Z8821N/ URL: https://www.fwo.be/). EV is supported by the Special Research Fund Belgium – Interdisciplinary Research Project (grant number: BOF.24J.2014.0005.02/ URL: https://www.ugent.be/en/research/funding/bof/iop). The funders had no role in study design, data collection and analysis, decision to publish, or preparation of the manuscript.”

Please provide an amended statement that declares *all* the funding or sources of support (whether external or internal to your organization) received during this study, as detailed online in our guide for authors at http://journals.plos.org/plosone/s/submit-now.

Please also include the statement “There was no additional external funding received for this study.” in your updated Funding Statement.

Author response # 2: The Funding Statement has been amended and included in the cover letter of this Response To Reviewers document: “SB is supported by the Research Foundation – Flanders; Belgium (FWO – 12Z8821N/ URL: https://www.fwo.be/). EV is supported by the Special Research Fund Belgium – Interdisciplinary Research Project (grant number: BOF.24J.2014.0005.02/ URL: https://www.ugent.be/en/research/funding/bof/iop). The funders had no role in study design, data collection and analysis, decision to publish, or preparation of the manuscript. There was no additional external funding received for this study.”

Academic editor:

Editor point # 1: What has previously been published on this topic and what does this work add to the existing literature?

Author response # 1: Thank you for this comment. There are different studies that focused on the association between mitochondrial dysfunctions and characteristics of ASD in general (for an overview see also meta-analyses of Rose et al. (2018) and Rossignol & Frye (2012)). Of interest is that developmental regression was noticed in 52% of the children with ASD who also had clear mitochondrial dysfunction, or with abnormal biochemical markers for mitochondrial function (Rossignol & Frye, 2012). This is much higher than in the general population of ASD where two meta-analyses estimated regression prevalence at 30% (Barger et al., 2013; Tan et al., 2021). Whether mitochondrial dysfunction contributed to or caused the reported regression in these children is unclear. An overview of the literature that has been published previously on the association between mitochondrial characteristics and characteristics of ASD, which was the starting point for our research questions, is included in the Introduction-section (p. 5-9). 

The present study aimed to evaluate lactate concentration in urine of children with ASD, as a non-invasive large-scale screening method for metabolic abnormalities including mitochondrial dysfunction, and its specific association with regression. To the best of our knowledge, we are the first to investigate this specific research question in a large community-based sample (which is less susceptible to referral bias than a clinical sample) with a non-invasive technique to measure metabolic characteristics. In addition, we also applied golden standard methods with regard to the measurement of ASD severity and developmental regression. This information is now included in the Introduction-section.

Adaptations in the revised manuscript – Introduction-section (p.8 lines 192-196 and p.9 lines 197-198): 

“The exploratory study presented here aimed to evaluate lactate concentration in urine of children with ASD, as a non-invasive large-scale screening method for metabolic changes characteristics including mitochondrial dysfunction, and its possible association with regression. To the best of our knowledge, the current study is the first to include (1) a large community-based sample, (2) a non-invasive technique to measure metabolic abnormalities, and (3) validated methods with regard to the measurement of ASD severity and developmental regression.”

After a new literature search, we came across one recent paper from the Richard E. Frye-group (Singh et al., 2020) that is also focussing on “Developmental regression and mitochondrial function in children with autism.”, however using a small clinical sample size (including a smaller age range), invasive technique (focusing on a unique type of mitochondrial dysfunction) and no validated methods to diagnose ASD and regression. The results from Singh et al. (2020) are now discussed in the Discussion-section.

Adaptations in the revised manuscript – Discussion-section (p.26 lines 531-539): “With regard to the prevalence of reported regression, no differences were found between the group of ASD children with low concentrations of lactate in urine and the group of ASD children with high concentrations of lactate in urine. Moreover, participants with ASD and regression (before or after 36 months) did not present a significantly higher concentration of lactate in urine as compared to ASD children without regression. These findings are in contrast with recent preliminary evidence for a unique type of mitochondrial dysfunction (i.e., abnormal respiratory function characterised by increased maximal oxygen consumption rate, maximal respiratory capacity and reserve capacity) which may be related to regression in a small clinical sample of children with ASD (72).” 

Next to the methodological aspects with regard to the large community-based sample size, non-invasive screening and highly validated diagnostic methods for both ASD and regression, our work adds to the existing literature that although children with ASD showed a broad range of phenotypic characteristics (that could fit into the diagnosis of mitochondrial disease or could be related to mitochondrial dysfunctions) we did not find an association between increased urinary lactate concentrations and regression. The present outcomes are in line with a previous comprehensive genetic study indicating that mitochondrial dysfunction is not a major contributing factor to the development of ASD. We can conclude that the evidence for this challenging association is still preliminary to date and that the role of mitochondrial dysfunctions in ASD is currently unclear. 

A non-invasive test to identify children with abnormalities consistent with mitochondrial dysfunction would enable researchers to better study and define mitochondrial dysfunction in ASD, eventually leading to treatment and better outcomes in these children. This information is also included throughout the Discussion-section of the manuscript (p. 25-31).

Editor point # 2: Is this a prospective study or a retrospective design? Research flow chart needs to be provided. How many patients included and excluded, why? How many patients lost in the follow-up or died in the discharge?

Editor point # 4: When and how the Urine samples were collected? Only one-time point was collected? How do Urine lactate concentration change over the course of follow-up or in the hospital?

Author response # 2 and # 4: Thank you for these two points including questions with regard to clarification on the methodology that was used. The current study is cross-sectional (only one measurement point was included) and history of developmental regression was measured retrospectively by parent report. This information is now added in the Participants’ information in the Method-section. We also included a recruitment and research flow chart in the Supporting Information (S1 File – see also below) to provide more information on the methodology of the present study.

Initially, 103 children were recruited from a community-based sample via social media, parent associations of children with ASD, home guidance organizations and different multidisciplinary rehabilitation centres through online and newsletter advertisements. However, in 3 children no urine samples were provided by the parents and in 1 child there was no official community diagnosis of ASD. Therefore these children (n = 4) were excluded from the present analyses. The final sample size contained 99 children between 3 and 11 years old. The questions on how many patients were lost in the follow-up or died in the discharge are not applicable to the current study.

Urine samples were collected at one time point. Given that we used the terminology ‘metabolic change’ in the title of the manuscript, we agree that this led to interpreting the research design in a prospective or longitudinal way. With ‘metabolic change’ we wanted to point to ‘abnormalities/deviations/defects in metabolic functioning’ in the children with ASD (with regression) as compared to children without ASD. Therefore, we decided to adapt the title and replace the word ‘change’ with ‘abnormalities’ throughout the manuscript. In this way, we hope that the purpose of the study and used design is more explicit.

All urine samples were collected postprandial (since in this way lactate concentration is more representative than fasting specimens; (Parikh et al., 2015) during the research setting or at home. The urine was collected through a urine bag (in very young children) or a urine sample beaker. Afterwards a volume of 11 mL was extracted from this sample and immediately frozen at a temperature of -18° Celsius before analysis within maximum 3 months after collection. In the lab, after thawing, the samples were immediately analysed. This information is now also included in the Procedure-overview in the Method-section. Information on developmental regression was collected at the same time point through retrospective parent-report (see also Method-section p.10-11).

Given that we included one measurement time point (i.e., cross-sectional study), it was not possible to analyse changes in urine lactate concentration. In the Implications of the Discussion-section (p.29-30) we included that “future research on mitochondrial dysfunction in ASD should focus on using large samples of infants and children with ASD, implementing serial measurements of different biomarkers. As lactate concentrations in urine fluctuate, serial measurements during several days would also be preferred above one single measurement.”

Adaptations in the revised manuscript - Participant information in Method-section(p. 9, lines 202-215): “This cross-sectional study was conducted in 99 children with ASD aged between 3 and 11 years old (M = 7.55 years, SD = 1.99; 72.7% boys). All children had an official community diagnosis of ASD confirmed by a multidisciplinary team fulfilling the criteria of the diagnosis of ASD according to the 4th (DSM-IV-TR; (43)) or 5th (DSM-5 (1)) edition of the Diagnostic and Statistical Manual of Mental Disorders (DSM). Initially, 103 children were recruited from a community-based sample via social media, parent associations of children with ASD, home guidance organizations and different multidisciplinary rehabilitation centres through online and newsletter advertisements. The primary purpose of the recruitment was to study the general and biological development of children with ASD (some children were also included in a related study (Boterberg et al., 2019)). However, in 3 children no urine samples were provided by the parents and in 1 child there was no official community diagnosis of ASD as defined above. Therefore these children (n = 4) were excluded from the present analyses. For an overview of the participant selection process see also the recruitment and research flow chart in the Supporting Information (S1 File). 

Adaptations in the revised manuscript - Measures information in Method-section (p. 10, lines 235-240): 

“History of Developmental regression.

The Autism Diagnostic Interview-Revised (ADI-R (47)) is a semi-structured, 111-item, diagnostic parent interview which is administered to classify ASD in children or adults. In the present study, information based on items #11-19 on history of language regression and items #20-28 on potential losses of other skills such as motor, self-help, play and social abilities was included.”

Adaptations in the revised manuscript - Procedure information in Method-section (p. 11, lines 268-269 and p. 12 line 270): “The urine collected through a urine bag (in young children) or a urine sample beaker was stored in test tubes with a volume of 11 mL and immediately frozen at a temperature of -18° Celsius before analysis within maximum 3 months after collection.”

Added Supplemental Information:

S1. Supporting Information on the recruitment and research flow chart (T1 = time point 1)

Editor point # 3: This study had been approved by the Ethics Committee？The approval no. of Ethics Committee should be listed and the informed consent was written and/ or oral?

Author response # 3: Thank you for this question. The current study was conducted under approval of both the ethical board of the Faculty of Psychology and Educational Sciences of the University Ghent and the University Hospital Ghent as conforming to the Declaration of Helsinki, 2000. We included the project numbers and Belgian registration number of the approval by the medical ethical committee in the revised manuscript. In addition, written informed consent was obtained from the parents or legal guardians of the children.

Adaptations in the revised manuscript – Method-section (p. 12, lines 271-276): “The study was conducted under approval of the ethical board of the Faculty of Psychology and Educational Sciences of the University Ghent (Project number: 2015/51; date: 03/08/2015) and the University Hospital Ghent (Belgian registration number: B670201525767; Project number: EC UZG 2015/1023; date: 27/10/2015) as conforming to the Declaration of Helsinki, 2000. In addition, written informed consent was obtained from the parents or legal guardians of the children.” 

Reviewer #1: Interesting research article describing null association between urinary lactate concentrations and autism. Discussion and limitations sections are informative even for describing negative results. Statistical analysis has been well performed.

Author response: Thank you for the positive feedback.

References

Barger, B. D., Campbell, J. M., & McDonough, J. D. (2013). Prevalence and onset of regression within autism spectrum disorders: a meta-analytic review. Journal of Autism and Developmental Disorders, 43(4), 817–828. https://doi.org/10.1007/s10803-012-1621-x

Boterberg, S., Van Coster, R., & Roeyers, H. (2019). Characteristics, Early Development and Outcome of Parent-Reported Regression in Autism Spectrum Disorder. Journal of Autism and Developmental Disorders, 49(11), 4603–4625. https://doi.org/10.1007/s10803-019-04183-x

Parikh, S., Goldstein, A., Koenig, M. K., Scaglia, F., Enns, G. M., Saneto, R., Anselm, I., Cohen, B. H., Falk, M. J., Greene, C., Gropman, A. L., Haas, R., Hirano, M., Morgan, P., Sims, K., Tarnopolsky, M., Van Hove, J. L. K., Wolfe, L., & DiMauro, S. (2015). Diagnosis and management of mitochondrial disease: a consensus statement from the Mitochondrial Medicine Society. Genetics in Medicine, 17(9), 689–701. https://doi.org/10.1038/gim.2014.177

Rose, S., Niyazov, D. M., Rossignol, D. A., Goldenthal, M., Kahler, S. G., & Frye, R. E. (2018). Clinical and molecular characteristics of mitochondrial dysfunction in autism spectrum disorder. Molecular Diagnosis & Therapy, 22(5), 571–593. https://doi.org/10.1007/s40291-018-0352-x

Rossignol, D. A., & Frye, R. E. (2012). Mitochondrial dysfunction in autism spectrum disorders: a systematic review and meta-analysis. Molecular Psychiatry, 17(3), 290–314. https://doi.org/10.1038/mp.2010.136

Singh, K., Singh, I. N., Diggins, E., Connors, S. L., Karim, M. A., Lee, D., Zimmerman, A. W., & Frye, R. E. (2020). Developmental regression and mitochondrial function in children with autism. Annals of Clinical and Translational Neurology, 7(5), 683–694. https://doi.org/10.1002/ACN3.51034

Tan, C., Frewer, V., Cox, G., Williams, K., & Ure, A. (2021). Prevalence and Age of Onset of Regression in Children with Autism Spectrum Disorder: A Systematic Review and Meta-analytical Update. Autism Research. https://doi.org/10.1002/aur.2463

---

## [Decision Letter · Decision Letter 1]

26 Aug 2022

Urine lactate concentration as a non-invasive screener for metabolic abnormalities: Findings in children with autism spectrum disorder and regression.

PONE-D-21-39785R1

Dear Dr. Boterberg,

We’re pleased to inform you that your manuscript has been judged scientifically suitable for publication and will be formally accepted for publication once it meets all outstanding technical requirements.

Kind regards,

Wen-Jun Tu

Academic Editor

PLOS ONE

Additional Editor Comments (optional):

Reviewers' comments:

Reviewer's Responses to Questions

**Comments to the Author**

1. If the authors have adequately addressed your comments raised in a previous round of review and you feel that this manuscript is now acceptable for publication, you may indicate that here to bypass the “Comments to the Author” section, enter your conflict of interest statement in the “Confidential to Editor” section, and submit your "Accept" recommendation.

Reviewer #1: All comments have been addressed

2. Is the manuscript technically sound, and do the data support the conclusions?

Reviewer #1: Yes

3. Has the statistical analysis been performed appropriately and rigorously? 

Reviewer #1: Yes

4. Have the authors made all data underlying the findings in their manuscript fully available?

Reviewer #1: Yes

5. Is the manuscript presented in an intelligible fashion and written in standard English?

Reviewer #1: Yes

6. Review Comments to the Author

Reviewer #1: Authors well answered to all the comments.

No more changes required.

This article can be accepted for publication

7. PLOS authors have the option to publish the peer review history of their article (what does this mean?). If published, this will include your full peer review and any attached files.

Reviewer #1: No

---

## [Editor Report · Acceptance letter]

31 Aug 2022

PONE-D-21-39785R1 

Urine lactate concentration as a non-invasive screener for metabolic abnormalities: Findings in children with autism spectrum disorder and regression. 

Dear Dr. Boterberg:

I'm pleased to inform you that your manuscript has been deemed suitable for publication in PLOS ONE. Congratulations! Your manuscript is now with our production department. 

Kind regards, 

on behalf of

Dr. Wen-Jun Tu 

Academic Editor

PLOS ONE